# Recent Advances in the Addition of Amide/Sulfonamide Bonds to Alkynes

**DOI:** 10.3390/molecules24010164

**Published:** 2019-01-04

**Authors:** Fei Zhao, Pinyi Li, Xiaoyan Liu, Xiuwen Jia, Jiang Wang, Hong Liu

**Affiliations:** 1Antibiotics Research and Re-evaluation Key Laboratory of Sichuan Province, Sichuan Industrial Institute of Antibiotics, Chengdu University, 168 Hua Guan Road, Chengdu 610052, China; zhaofei@cdu.edu.cn (F.Z.); pinyiLi19950206@126.com (P.L.); 19940826097@163.com (X.L.); jiaxiuwen2018@126.com (X.J.); 2State Key Laboratory of Drug Research and CAS Key Laboratory of Receptor Research, Shanghai Institute of Materia Medica, Chinese Academy of Sciences, 555 Zuchongzhi Road, Shanghai 201203, China; jwang@simm.ac.cn; 3University of Chinese Academy of Sciences, No.19A Yuquan Road, Beijing 100049, China

**Keywords:** amide bond, sulfonamide bond, alkynes, addition reaction, aminoacylation, aminosulfonylation

## Abstract

The addition of amide/sulfonamide bonds to alkynes is not only one of the most important strategies for the direct functionalization of carbon–carbon triple bonds, but also a powerful tool for the downstream transformations of amides/sulfonamides. The present review provides a comprehensive summary of amide/sulfonamide bond addition to alkynes, including direct and metal-free aminoacylation, based-promoted aminoacylation, transition-metal-catalyzed aminoacylation, organocatalytic aminoacylation and transition-metal-catalyzed aminosulfonylation of alkynes up to December 2018. The reaction conditions, regio- and stereoselectivities, and mechanisms are discussed and summarized in detail.

## 1. Introduction

The addition of atom–atom bonds to alkynes has become an important strategy for the functionalization of carbon–carbon triple bonds [1,2,3,4,5,6,7,8,9,10,11,12,13,14,15,16]. These intermolecular and intramolecular addition reactions provide a facile and efficient access to highly functionalized alkenes and cyclic compounds, respectively, in a high atom- and step-economic manner. Considering the large occurrence of amide/sulfonamide motifs in natural products and pharmaceutical agents, the addition of amide/sulfonamide bonds to alkynes, namely aminoacylation/aminosulfonylation of alkynes, is particularly important. Because they allow the direct downstream transformations of amides/sulfonamides by the insertion of carbon–carbon triple bonds into the amide/sulfonamide bonds, they thus produce more complex and skeletally different addition molecules (Scheme 1). In addition, the aminoacylation/aminosulfonylation of alkynes also constitutes a tool for the structural modification of compounds carrying amide/sulfonamide bonds, especially for peptides, which are an important class of drugs used in the clinic [17,18,19]. Besides, amide/sulfonamide bond addition to alkynes, which constructs one C–C/S and one C–N bond in a single step featuring high atom- and step-economy, is in accordance with the concept of “green and sustainable chemistry”.

It should be noted that the addition of amide/sulfonamide bonds to alkynes has not been reviewed before. Moreover, amide/sulfonamide bond addition to alkynes has achieved many important developments in recent decades, especially in transition-metal-catalyzed and organocatalytic processes. Therefore, a review focused on the aminoacylation/aminosulfonylation of alkynes would enrich the knowledge of synthetic chemists who are interested in amide/sulfonamide bond activation. The aim of the present review is to provide a systematical and comprehensive summary on the addition of amide/sulfonamide bonds to alkynes, including direct and catalyst-free aminoacylation, based-promoted aminoacylation, transition-metal-catalyzed aminoacylation, organocatalytic aminoacylation, and transition-metal-catalyzed aminosulfonylation of alkynes up to December 2018. We hope this review will serve as a handy reference for chemists interested in the addition of amide/sulfonamide bonds to alkynes, and will encourage further developments in this field in overcoming the remaining challenges.

## 2. Addition of Amide Bonds to Alkynes

### 2.1. Direct Addition of Amide Bonds to Alkynes without Catalysts and Additives

The first example of amide bond addition to alkynes was a catalyst- and additive-free process as reported by Eğe’s group in 1976 [20]. They found the treatment of active 2-phenylpyrazolidin-3-ones **4** with dimethyl acetylenedicarboxylate **5** in CH_3_CN under reflux led to the formation of the interesting ring expansion products 1,2-diazepin-5-ones **6**, albeit with unsatisfactory selectivities and yields (Scheme 2). The main byproducts of this reaction were the *cis*- and *trans*- Michael-type addition products. Besides, this transformation was strongly influenced by the solvent used. Polar but nonprotic solvents such as acetone and acetonitrile gave the best results while few products were obtained in protic solvents such as ethanol. Similar results were also observed in Svete and Stanovnik’s research on the addition reactions between 5,5-dimethyl-2-(1*H*-indenyl-2)-3-pyrazolidinones **7** and acetylenedicarboxylates **8** (Scheme 3) [21]. A plausible reaction mechanism was outlined in Scheme 4. The Michael addition of N1 of the pyrazolidinones to acetylenedicarboxylate generates the carbanionic intermediate **11**, which attacks the carbonyl group across the ring to give the bicyclic amino-acetal intermediate **12**. The following ring opening of **12** affords the zwitterionic intermediate **13**, which undergoes ring expansion to produce the addition products **6**.

In contrast, Hanack’s group developed a more general and practical addition of amide bonds to carbon–carbon triple bonds without any catalysts and additives in 1989 [22]. Amides **15** were directly added to alkynyl trifluoromethyl sulfones **14** to afford the *cis*-adducts **16** with excellent regioselectivity and good yields, despite the fact that a long reaction time was required (Scheme 5). This protocol showed advantages such as simple operation, broad substrate scope, and the avoidance of metals and additives. A plausible mechanism was proposed in Scheme 6. The Michael reaction of nitrogen atom of the amides to alkynyl trifluoromethyl sulfones yields a zwitterion **17**, which undergoes cyclization to form intermediate **18**. The subsequent rupture of the carbon–nitrogen bond of **18** gives the products **16**. The regio- and stereoselectivity observed in this reaction could be well explained by this mechanism.

What seems particularly interesting is the addition of 1,1′-carbonyldiimidazole (CDI) **19** to alkynoic acids **20** with the release of CO_2_, as reported by Knölker and co-workers in 1993 (Scheme 7) [23]. This reaction proceeded well under mild conditions to provide the *E*-adducts **23** in moderate yields. The reaction of CDI **19** with alkynoic acids **20** generated intermediate **21** and imidazole **22** with the release of CO_2_, and the subsequent addition of the imidazole **22** to the electron deficient alkyne **21** stereoselectively produced the products **23**.

### 2.2. Base-Promoted Addition of Amide Bonds to Alkynes

In 1987, Suzuki and Tsuchihashi disclosed a sequential process for the preparation of enaminones **27** through the insertion of lithium (triphenylsilyl)acetylide into amides (Scheme 8) [24]. Acyclic amides reacted smoothly to give the *E*-enaminones in high yields, while lower yields of the desired ring expansion products were obtained when cyclic amides were used as the substrates. It is worth noting that the triphenyl group on silicon was essential for the transformation as other silylacetylides failed to give the enaminone products. The possible reaction pathway may involve the initial formation of the silylalkynone, the subsequent Michael addition of in situ-formed lithium amide and the final protiodesilylation.

Subsequently, Jeong et al. successfully realized the addition of Weinreb amides **29** to the carbon–carbon triple bond of trifluoropropynyl lithium **30** in a one-pot two-step pathway [25,26,27], providing a *Z*/*E* mixture of β-trifluoromethyl enaminones **34** in moderate yields (Scheme 9). It should be noticed that an excessive amount of trifluoropropynyl lithium was required to consume Weinreb amides completely. The reaction temperature had a decisive impact on the outcome of this transformation since quenching the reaction with H_2_O at room temperature failed to give the enaminones but gave recovery of Weinreb amides. Besides, the use of *N*,*N*-dimethylbenzamide instead of *N*-methoxy-*N*-methylbenzamide under optimal conditions did not provide the desired product at all, only the recovery of starting material. This indicated that the *N*-methoxy group in Weinreb amides played an indispensable role in this reaction. The proposed mechanism involved the key intermediate **31**, which was formed from the addition of trifluoropropynyl lithium with Weinreb amides. Then **31** was quenched by H_2_O to give the ynone intermediate **32**, which rapidly reacted with *N*-methoxy-*N*-methylamine **33** generated from the reaction to give the products **34**. The *N*-methoxy group in Weinreb amides was essential because the oxygen could coordinate with the lithium cation to stabilize the key intermediate **31**. Particularly, the fact that trapping **31** with trimethylsilyl chloride afforded the corresponding siloxane derivative in a high yield further demonstrated the mechanism.

Soon afterwards, the group of Nielsen reported the insertion of sodium acetylide of ethyl propynoate **35** into Weinreb amides **29** to produce the 1,2-addition products **37** as the major products (Scheme 10a) [28]. The selectivity of the 1,2-addition products **37** over 1,1-addition products **38** depended on the R group of the Weinreb amides. Substrates with bigger R substituents showed higher selectivity than those with smaller ones. For example, substituents such as phenyl showed excellent selectivity, providing the 1,2-addition adduct as the single product in high yield. However, substrates carrying bulky substituents such as tert-butyl or 2,4-dimethoxyphenyl did not undergo this transformation. Notably, the tertiary enaminones **37** preferentially adopted *E*-geometry in all cases, suggesting the 1,2-addition reactions proceeded in a highly trans-selective manner. In addition, the β-enaminoketoesters **37** were employed by the authors to react with hydrazines **39** under microwave irradiation to construct pyrazoles **40** through a regioselective cyclocondensation (Scheme 10b). Similarly, Choudhury’s group reported a one-pot sequential process consisting of nucleophilic substitution of the lithiated acetylides with Weinreb amides, and a following Michael reaction of the extruded *N*-methoxy-*N*-methylamine to a carbon–carbon triple bond after quenching with saturated NH_4_Cl, producing the *E*-β-enamino ketones **43** as the single geometrical isomer in high yields (Scheme 11) [29].

Very recently, Li and co-workers reported the addition of the amide bond of imides **45** to the carbon–carbon triple bond of alkynones **44** under basic conditions (Scheme 12) [30]. This addition reaction proceeded smoothly with the addition of a base such as K_2_CO_3_ in DMSO at high temperature, affording the corresponding tetra-substituted enamides **46** in good yields. Although this transformation suffered from unsatisfactory stereoselectivities, excellent regioselectivities were observed. The acyl group and amide group were dominantly located at the α-position and β-position of the carbonyl, respectively. Interestingly, in the reactions of alkynones **47** bearing an *ortho*-bromo-substituted aryl ring, highly functional chromones **48** were selectively formed in good to high yields via the *O*-cyclization pathway (Scheme 13). Control experiments showed that the base played an important role. It could deprotonate the imides **45** to form a nitrogen anion, which undergoes a Michael-type addition to the alkynones **49** to produce the anion intermediate **50**. Then intermediate **50** undergoes an intramolecular nucleophilic addition/ring-opening sequence to provide intermediate **52**. Hydrolysis of intermediate **52** generates enamides **46** (X = H), or imine–enamine tautomerization of intermediate **52** followed by nucleophilic aromatic substitution (S_N_Ar) to give the chromones **48** (X = Br) (Scheme 14).

Soon afterwards, Li and co-workers presented the insertion of alkynones **53** into the amide bond of amide **54** promoted by Cs_2_CO_3_ (Scheme 15), providing the functionalized enaminones **55** with high stereoselectivity and excellent regioselectivity [31]. It should be noted that the combination of Cs_2_CO_3_ and 1,10-phenanthroline hydrate (Phen∙H_2_O) is essential to obtain good yields in a short reaction time. The authors hypothesized that 1,10-phenanthroline hydrate may act as a metal ion chelator, which increased the basicity of Cs_2_CO_3_ to accelerate the reaction. Similarly, 3-carbonyl-4-quinolinones **58** were selectively formed via a subsequent *N*-cyclization pathway in the cases of alkynones **56** bearing an *ortho*-bromo-substituted aryl ring (Scheme 16). The proposed reaction mechanism is outlined in Scheme 17. The Michael-type addition of amides **57** to alkynones **59** under basic conditions yields an allenol intermediate **60**. The subsequent intramolecular nucleophilic addition gives a highly reactive cyclobutenol intermediate **61**, which undergoes ring opening to produce a formal alkyne insertion intermediate **62**. Then intermediate **62** undergoes imine-enamine tautomerization to provide intermediate **63**, which undergoes a nucleophilic aromatic substitution (S_N_Ar) to afford the quinolinone products **64** (Y = Br). In contrast, the protonation of intermediate **62** or **63** leads to the formation of the enaminone products **65** (Y = H).

### 2.3. Transition-Metal-Catalyzed Addition of Amide Bonds to Alkynes

In 2004, Yamamoto’s group reported the platinum catalyzed synthesis of highly functional indoles through an intramolecular amide C–N bond addition to alkynes with the [1,3]-migration of acyl groups (Scheme 18) [32]. PtCl_2_ showed the highest catalytic activity compared with other platinum catalysts such as PtCl_2_(CH_3_CN)_2_, PtBr_2,_ and Pt(PPh_3_)_4_. Various *ortho*-alkynylanilides **66** bearing diverse alkyl or aryl groups at R_1_ could be converted into the corresponding indole products **67** with good to high yields with PtCl_2_. Notably, a variety of acyl groups could undergo intramolecular [1,3]-migration to give 3-acyl-indoles **67**. The main drawback of this method is that the desired products **67** are together with deacylated byproducts **68** in most cases. Based on the results of deuterium-labeling experiments and crossover experiments, the authors proposed a catalytic cycle of this intramolecular aminoacylation of alkynes. As shown in Scheme 19, coordination of alkyne moiety to PtCl_2_ yields the π-complex **69**, followed by nucleophilic attack of nitrogen to the alkyne, affording the zwitterionic intermediate **70**. An intramolecular [1,3]-migration of the acyl group then gives intermediate **71**, which affords the product and regenerates the catalyst. The 3-deacylated byproducts **68** may attribute to the deacylation which takes place through the protonolysis of the C–Pt bond of intermediate **70**.

Encouraged by the excellent catalytic performance of platinum catalysts towards the intramolecular aminoacylation of alkynes, Nakamura’s group further studied similar reactions using *ortho*-alkynylphenylureas or *ortho*-alkynylphenyl carbamates as substrates (Scheme 20) [33]. The reactions of *ortho*-alkynylphenylureas **72** having a carbamoyl group attached to the nitrogen atom proceeded successfully under the catalysis of PtI_4_, providing the desired indole-3-carbamides **73** in moderate to high yields along with the 3-protonated byproducts **68**. Interestingly, *ortho*-alkynylphenyl carbamates **74** could be converted into the corresponding indole-3-carboxylates **75** in good yields without the generation of 3-protonated byproducts **76**. The authors proposed a similar mechanism to that of Yamamoto’s group [32]. They assumed the generation of the 3-protonated byproducts **68** in the reactions of *ortho*-alkynylphenylureas **72** may be attributed to protodemetalation of intermediate **78** by a proton from the methyl moiety of intermediate **79**, which was extruded in the reaction (Scheme 21). Notably, this work proved that amide and ester groups could be used as the migrating groups, thus providing an efficient method to synthesize indole-3-carbamides/carboxylates which could not be prepared via Friedel–Crafts electrophilic substitution into the C3-position of the indole ring.

In 2007, Nakamura’s group revealed that PdBr_2_ could also catalyze the intramolecular amide C–N bond addition to alkynes (Scheme 22), affording the indole adduct **82** from *ortho*-alkynylanilide **81** in 52% yield [34]. Encouraged by the catalytic performance of PdBr_2_ towards the intramolecular aminoacylation of alkynes, Liu’s group further screened a series of palladium complexes. They found that PdCl_2_(CH_3_CN)_2_ showed excellent catalytic activity (Scheme 23) [35]. Substrates **83** with alkyl/aryl groups at R_1_ furnished the corresponding products **84** in good to excellent yields. The protocol was also compatible with substrates **83** bearing electron-donating substituents, halides, and electron-withdrawing substituents at R_2_, which produced the corresponding products **84** in high yields. In addition, the reactions of substrates **83** with different alkyl substituents at R_3_ also took place smoothly, providing the desired products **84** in high yields. More importantly, various acyl and amide groups could migrate smoothly and be conveniently introduced at the C3-position of indoles.

Subsequently, Liu and coworkers further extended their palladium catalytic system to the synthesis of 3-diketoindoles **86** from *ortho*-alkynyl-*N*-α-ketoacylanilines **85** via the intramolecular amide bond addition to alkynes (Scheme 24) [36]. Notably, this addition reaction proceeded smoothly to give the high functional 3-diketoindoles **86** with the [1,3]-migration of α-ketoacyl groups, which were used as migrating groups for the first time. Compared with previously reported protocols such as Friedel–Crafts acylation [37], Glyoxylation/Stephens–Castro coupling sequence [38], and the oxidative cross-coupling of indoles [39,40,41], which achieved the synthesis of 3-diketoindoles through the modification of the indole ring, but suffered from poor selectivity, operational complexity, the requirement of strict exclusion of moisture, limited substrate scope and low atom economy, this new method successfully prepared 3-diketoindoles via the construction of an indole ring with valuable features such as operational simplicity, high atom economy, broad substrate scope and high yields. Interestingly, a 3-diketoindole dimer **88** was synthesized in a high yield when substrate **87** was subjected to the optimal reaction conditions (Scheme 25). Finally, the authors proposed a reaction mechanism which is similar to that proposed by Yamamoto’s group [32].

Ruthenium complexes were also found to be efficient catalysts for the intramolecular amide bond addition to alkynes, as was reported by Li’s group in 2012 (Scheme 26) [42]. Their study showed that [RuCl_2_(*p*-cym)]_2_ displayed the highest catalytic activity, with which *ortho*-alkynylanilides **83** could undergo intramolecular annulation through amide bond addition to the alkyne moiety to synthesize highly functional indoles **84**. A variety of substrates **83** carrying diverse functional groups such as olefin, ester, aldehyde were well tolerated and could be converted into the corresponding indole products **84** in moderate to high yields. Despite the fact that a longer reaction time was required compared with platinum or palladium catalytic systems, it is worth noting that no 3-deacylated indoles were observed in all examples. However, the main shortcoming of this method is that unsatisfactory yields were obtained when bigger acyl groups such as acetyl were employed as the migrating groups. Based on the mechanistic study results with deuterium-labeling experiments, the authors hypothesized that the reaction mechanism may involve the complexation of substrates with ruthenium catalyst, the subsequent oxidative addition of ruthenium catalyst across the amide bond, the following addition of the N–Ru bond to carbon–carbon triple bonds, and the final reductive elimination to produce the products and regenerate the catalyst (Scheme 27).

### 2.4. Addition of Amide Bonds to Alkynes through Organocatalysis

In addition to the metal-catalyzed processes, methods utilizing organocatalysis, which feature advantages such as low cost, environmental economy, and the avoidance of metal contamination, have also been developed in recent years. What seems particularly interesting is the insertion of an electron-deficient alkyne **5** into the amide bond of an acyl-onio salt **92** (Scheme 28), as reported by Weiss and Huber [43]. This reaction could be achieved in the presence of a catalytic amount of small organic molecules such as PPh_3_ or DMAP, providing the desired β-oniovinylation products **93** in good yields. The stereochemistry of this process depends on the reaction conditions, preferentially *E*- or *Z*-stereochemistry was observed, and the *Z*-isomer is the thermodynamically more stable isomer. More importantly, the onio substituent in the products **93** could be selectively replaced by a number of nucleophiles, such as anilines, phenols, and thiophenols, to prepare Michael systems with donor functions in the β-position, which could be further converted into quinolones, thiochromones, and pyrazoles by intramolecular cyclization. The authors proposed a catalytic cycle for this organocatalytic process. Taking the transformation catalyzed by PPh_3_ as the example (Scheme 29), the conjugate addition of PPh_3_ to alkyne **5** produces the zwitterionic intermediate **94**, which attacks the electrophilic carbonyl center of **92** to provide intermediate **95** with liberation of 4-dimethylaminopyridine (DMAP). Then intermediate **95** reacts with the liberated DMAP to give the products **93** and regenerates the catalyst.

Recently, Doi’s group reported an example of amide addition to alkynes through tertiary amine organocatalysis (Scheme 30) [44]. They found that *o*-alkynoylaniline derivatives **96** could undergo intramolecular aminoacylation of the carbon–carbon triple bonds successfully under the catalysis of 9-azajulolidine (9-AJ) to afford the 3-acyl-4-quinolinones **97** in moderate to good yields with excellent regioselectivity. Notably, a variety of acyl groups including ester groups could act as migrating groups to be transferred to the C3-positon of the quinolinones. Particularly, the synthesis of pyrrolyl 4-quinolinone alkaloid, quinolactacide, and its analogues were successfully achieved by the authors employing this organocatalytic process. Finally, a plausible reaction mechanism was outlined in Scheme 31. 1,4-addition of 9-AJ to substrates **96** takes place at first, and the subsequent nucleophilic attack of the resulting anion to the acyl group provides intermediate **98**, which could be converted into the intermediate **99**. Then the acyl group in allenolate **99** could be transferred to the C3-position, thus leading to the formation of enone **100**, which undergoes 6-endo cyclization to provide the products **97** with the regeneration of 9-AJ.

## 3. Transition-Metal-Catalyzed Addition of Sulfonamide Bonds to Alkynes

The group of Nakamura reported the first example of the addition of sulfonamides to alkynes in 2007. As shown in Scheme 32, *ortho*-alkynyl-*N*-sulfonylanilines **101** could undergo the intramolecular aminosulfonylation of carbon–carbon triple bonds successfully under the catalysis of AuBr_3_ to give the 3-sulfonylindoles **102** in good to high yields [45,46]. Although small amounts of 4- and 6-sulfonylindoles were obtained as the byproducts in some examples, this process provides a facile and efficient method for the synthesis of 3-sulfonylindoles, which cannot be synthesized directly from the corresponding unsubstituted indoles by electrophilic substitution because the electrophilicity of the sulfonyl groups is much lower than that of the acyl groups and halogens [47,48]. Interestingly, when 2-alkynyl-6-methoxy-*N*-sulfonylanilines **105** were employed as the substrates, this transformation could also occur under the catalysis of InBr_3_. However, the intramolecular aminosulfonylation products **106** were obtained as minor products, whereas 6-sulfonylindoles **107** were observed as the major products (Scheme 33). The reaction mechanism of this process may involve the initial coordination of the catalyst to the alkyne moiety, subsequent nucleophilic attack of the nitrogen atom to the carbon–carbon triple bond, the following migration of the sulfonyl group to the C3-position of the indole skeleton, and the final generation of the products with elimination of the catalyst (Scheme 34).

Subsequently, Chan’s group presented a gold-catalyzed domino aminocyclization/1,3-sulfonyl migration of *N*-substituted *N*-sulfonyl-aminobut-3-yn-2-ols **111** to synthesize highly functional pyrroles **112** (Scheme 35) [49]. A screening of gold catalysts disclosed that the NHC (*N*-heterocyclic carbene)-gold(I) complex **113** was found to be the most effective catalyst for this intramolecular aminosulfonylation of alkynes. It could catalyze the conversion of various substrates **111** carrying electron-withdrawing, electron-donating, and sterically demanding groups to the corresponding pyrroles **112** in moderate to high yields. It is worth noting that this method provides an efficient and convenient tool to prepare penta-substituted highly functional pyrroles. The mechanism of this transformation is outlined in Scheme 36. The coordination of gold cation to the alkyne moiety of the substrates gives intermediate **114**, which undergoes a nucleophilic attack of the nitrogen atom to the alkynes to produce intermediate **115**. At this juncture, dehydration of intermediate **115** yields cationic pyrrole–gold adduct **116**, subsequent 1,3-sulfonyl migration of intermediate **116** then results in deauration with the regeneration of the gold catalyst and delivery of the products **112**. Alternatively, intermediate **115** could undergo the deaurative 1,3-sulfonyl migration first to afford intermediate **117**, which undergoes dehydrative aromatization to produce the products **112**.

Soon afterwards, Liu’s group reported a more general and efficient intramolecular aminosulfonylation of alkynes to synthesize 3-sulfonylindoles **119** through palladium catalysis (Scheme 37) [35,50]. The reactions took place smoothly with PdCl_2_(CH_3_CN)_2_ in CH_3_CN at 90 °C to afford 3-sulfonylindoles **119** as the single products without the generation of 4- and 6-sulfonylindole regioisomers, which were obtained as the byproducts in the gold- and indium-catalyzed process. In addition, this protocol features broad substrate scope, good functional group tolerance, and moderate to high yields, thus providing a practical access to functional 3-sulfonylindoles. A plausible mechanism, which is similar to gold- or indium-catalyzed aminosulfonylation of alkynes [45,46], was also proposed by the authors.

## 4. Conclusions and Perspectives

In this review, we presented a summary of the addition of amide/sulfonamide bonds to alkynes, which has emerged as a highly important tool to functionalize carbon–carbon triple bonds. The aminoacylation/aminosulfonylation of alkynes, which is characterized by high atom- and step-economy in an environmentally-friendly manner, has also become a remarkable method for the downstream transformations of amide/sulfonamides compounds. Notably, the intramolecular aminoacylation/aminosulfonylation of alkynes has provided a facile and efficient protocol for the synthesis of valuable heterocycles such as chromones, quinolinones, indoles, and pyrroles. Despite the remarkable achievements made, there are at least three areas where some critical advances are necessary to make the aminoacylation/aminosulfonylation of alkynes more general and powerful: (a) the intermolecular addition of unactivated amide/sulfonamide bonds to alkynes is still in high demand; (b) as the reported metal-catalyzed process employed expensive metals, the development of cheap metal catalysis as well as organocatalysis, will be a good direction to take; (c) the exploration of tandem reactions involving aminoacylation/aminosulfonylation of alkynes will continue to drive this field considering their efficiency and step-economy in constructing complex heterocyclic compounds.

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
