# Peer review of "Recent Advances in the Addition of Amide/Sulfonamide Bonds to Alkynes"

_molecules, 2019, doi:10.3390/molecules24010164_

Round 1
Reviewer 1 Report
Liu and co-workers presented in their review article the current state of knowledge about the addition of amides and sulfonamides to alkynes. Researchers provide a systematical and comprehensive summary of these type reactions, including direct and catalyst-free aminoacylation, based-promoted aminoacylation, transition-metal-catalyzed aminoacylation, organocatalytic aminoacylation and transition-metal-catalyzed aminosulfonylation of alkynes. Manuscript very well written, with good examples, therefore I recommend this article for publication in Molecules in present form.
Author Response
Reviewer 1
Comments: Liu and co-workers presented in their review article the current state of knowledge about the addition of amides and sulfonamides to alkynes. Researchers provide a systematical and comprehensive summary of these type reactions, including direct and catalyst-free aminoacylation, based-promoted aminoacylation, transition-metal-catalyzed aminoacylation, organocatalytic aminoacylation and transition-metal-catalyzed aminosulfonylation of alkynes. Manuscript very well written, with good examples, therefore I recommend this article for publication in Molecules in present form.
Answer: Thanks a lot for your positive comments and recommendation. The aim of the present review is to provide a systematical and comprehensive summary on the addition of amide/sulfonamide bonds to alkynes. We hope this review will serve as a handy reference for chemists interested in the addition of amide/sulfonamide bonds to alkynes.
Reviewer 2 Report
This paper summarized development of coupling reactions of amides and sulfonamides with alkynes in the presence and absence of catalysts, such as metal catalysts and organocatalysts. This reaction can provide unsaturated carbonyl compounds and indole derivatives that are important compounds in organic synthesis. The scopes and limitations of substrates and reaction mechanism are clear. This article is interesting and useful for readers in the field of organic chemistry. This referee accepts the publication in Molecules.
Author Response
Comments: This paper summarized development of coupling reactions of amides and sulfonamides with alkynes in the presence and absence of catalysts, such as metal catalysts and organocatalysts. This reaction can provide unsaturated carbonyl compounds and indole derivatives that are important compounds in organic synthesis. The scopes and limitations of substrates and reaction mechanism are clear. This article is interesting and useful for readers in the field of organic chemistry. This referee accepts the publication in Molecules.
Answer: Thanks so much for your positive comments and recommendation. The addition of atom-atom bonds to alkynes has become an important strategy for the functionalization of carbon–carbon triple bonds. These intermolecular and intramolecular addition reactions provide a facile and efficient access to highly functionalized alkenes and cyclic compounds, respectively, in a high atom- and step-economic manner. We hope this review will serve as a handy reference for chemists interested in the addition of amide/sulfonamide bonds to alkynes, and will encourage further developments in this field to overcome remaining challenges.
Reviewer 3 Report
This review shows the development of the addition of amide/sulfonamide bonds to alkynes in past decades, including metal-free, base-promoted and transition-metal-catalyzed reaction pathways. The author concluded most of the major reactions published on this topic and explained the corresponding mechanisms, which showed solid knowledge background and pretty good understanding in this field. Therefore, this reviewer would recommend its publishing in Molecules after addressing the following issues.
1. Most of the part in this review is pretty substantial. However, part 2.4 “Addition of amide bonds to alkynes through organocatalysis” only have two published works included. It would be much better to add at least one more paper to this chapter.
2. Please use representative substrates while introducing reactions but not the whole substrate scope.
3. Several mechanisms are almost identical to each other. The author should only draw and introduce the first one in detail, then simply cite it when facing the same mechanisms. (For example, Scheme 19, 21 and 27; Scheme 36 and 40)
4. The overall language should be refined. Too much unnecessary clause used.
5. There are several minor mistakes in this review:
1) Line 27 “element-element bonds” better described as “atom-atom bonds” or more specifically “C-N/C-S bonds”.
2) Line 46 Scheme 1. The double bond in product 3 looks weird.
3) Line 79 – 80 Add “was” between “Amides 15” and “added to”.
4) Line 80 – 82 Phrase “afford the … in good yields” can be rewrite as “afford the cis-adducts 16 with excellent regioselectivity and good yields”.
5) Line 93 and Line 306 “Particularly interesting is” should be “What seems particularly interesting is”.
6) Line 97 delete “moiety of”.
7) Line 115 change “manner” to “pathway”.
8) Line 117 and 177 “noted” should be “noticed”.
9) Line 122 “, this” change to “. This”.
10) Line 129 delete “of”.
11) Line 133 “represented” should be “reported”.
12) Line 134 – 135 all the “adducts” should be “products”. (because you have “addition” in front of it)
13) Line 156 “aid” should be “add”.
14) Line 163 “a crucial and indispensable role, which” change to “an important role. It”.
15) Line 165 “, intermediate 50 then” should be “. Then intermediate 50”.
16) Line 178 delete “use”.
17) Line 256 add “the” in front of “synthesis”.
18) Line 266 change “of” to “such as”.
19) Line 288 “83” should be “88”.
20) Scheme 29. The intermediate 98 cannot convert to 99. If Ru work as π acid and activate the triple bond, the next step should be nucleophilic attack to the triple bond. If the next step is oxidative addition to the amide bond, then Ru should coordinate with N or O first.
21) Line 352 “, however” should be “. However”.
22) Line 368 “carbine” should be “carbene”.
23) Line 380 and 381 “depuration” should be “deauration”.
Author Response
Thanks a lot for the reviewer’s comments, and we answer the comments one by one as follows.
Comment 1: Most of the part in this review is pretty substantial. However, part 2.4 “Addition of amide bonds to alkynes through organocatalysis” only have two published works included. It would be much better to add at least one more paper to this chapter.
Answer: Thanks a lot for your good suggestion. We have done a systematic literature survey again using the professional software “SciFinder” and tried our best to find more literatures on amide bond addition to alkynes through organocatalysis. But we found that there are only two related examples so far, which have already been summarized in our review. We feel sorry about that, but we believe there will be more reports on amide bond addition to alkynes through organocatalysis in the future, and we are happy to renew our review at that time.
Comment 2: Please use representative substrates while introducing reactions but not the whole substrate scope.
Answer: Thanks so much for your good suggestion. We have made the corresponding changes in our revised manuscript according to your suggestion, and only several representative substrates were used to illustrate the reactions. For your information, most of the reaction schemes have been revised, and more details can be found in our revised manuscript.
Comment 3: Several mechanisms are almost identical to each other. The author should only draw and introduce the first one in detail, then simply cite it when facing the same mechanisms. (For example, Scheme 19, 21 and 27; Scheme 36 and 40)
Answer: Thanks so much for indication and good suggestion. We have deleted the original Scheme 21, 27 and 40, and made the corresponding changes according to you suggestion. Specifically, we cite the original literature when facing the similar reaction mechanisms. For your information, more details can be found in our revised manuscript.
Comment 4: The overall language should be refined. Too much unnecessary clause used.
Answer: Thanks so much for your good suggestion. We have carefully refined the language and reduced the unnecessary clause. For your information, more details can be found in our revised manuscript. Thanks a lot for your comments which can help us to improve the quality of our manuscript
Comment 5: There are several minor mistakes in this review: 1) Line 27 “element-element bonds” better described as “atom-atom bonds” or more specifically “C-N/C-S bonds”.
Answer: Thanks so much for your indication, and we have made the corresponding changes according to your suggestion.
Comment 6: 2) Line 46 Scheme 1. The double bond in product 3 looks weird.
Answer: Thanks so much for your indication, and we have made the corresponding changes according to your suggestion.
Comment 7: 3) Line 79-80 Add “was” between “Amides 15” and “added to”.
Answer: Thanks so much for your indication, and we have made the corresponding changes according to your suggestion.
Comment 8: 4) Line 80-82 Phrase “afford the … in good yields” can be rewrite as “afford the cis-adducts 16 with excellent regioselectivity and good yields”.
Answer: Thanks so much for your indication, and we have made the corresponding changes according to your suggestion.
Comment 9: 5) Line 93 and Line 306 “Particularly interesting is” should be “What seems particularly interesting is”.
Answer: Thanks so much for your indication, and we have made the corresponding changes according to your suggestion.
Comment 10: 6) Line 97 delete “moiety of”.
Answer: Thanks so much for your indication, and we have made the corresponding changes according to your suggestion.
Comment 11: 7) Line 115 change “manner” to “pathway”.
Answer: Thanks so much for your indication, and we have made the corresponding changes according to your suggestion.
Comment 12: 8) Line 117 and 177 “noted” should be “noticed”.
Answer: Thanks so much for your indication, and we have made the corresponding changes according to your suggestion.
Comment 13: 9) Line 122 “, this” change to “. This”.
Answer: Thanks so much for your indication, and we have made the corresponding changes according to your suggestion.
Comment 14: 10) Line 129 delete “of”.
Answer: Thanks so much for your indication, and we have made the corresponding changes according to your suggestion.
Comment 15: 11) Line 133 “represented” should be “reported”.
Answer: Thanks so much for your indication, and we have made the corresponding changes according to your suggestion.
Comment 16: 12) Line 134-135 all the “adducts” should be “products”. (because you have “addition” in front of it)
Answer: Thanks so much for your indication, and we have made the corresponding changes according to your suggestion.
Comment 17: 13) Line 156 “aid” should be “add”.
Answer: Thanks so much for your indication, and we have made the corresponding changes according to your suggestion.
Comment 18: 14) Line 163 “a crucial and indispensable role, which” change to “an important role. It”.
Answer: Thanks so much for your indication, and we have made the corresponding changes according to your suggestion.
Comment 19: 15) Line 165 “, intermediate 50 then” should be “. Then intermediate 50”.
Answer: Thanks so much for your indication, and we have made the corresponding changes according to your suggestion.
Comment 20: 16) Line 178 delete “use”.
Answer: Thanks so much for your indication, and we have made the corresponding changes according to your suggestion.
Comment 21: 17) Line 256 add “the” in front of “synthesis”.
Answer: Thanks so much for your indication, and we have made the corresponding changes according to your suggestion.
Comment 22: 18) Line 266 change “of” to “such as”.
Answer: Thanks so much for your indication, and we have made the corresponding changes according to your suggestion.
Comment 23: 19) Line 288 “83” should be “88”.
Answer: Thanks so much for your indication, and we have made the corresponding changes according to your suggestion.
Comment 24: 20) Scheme 29. The intermediate 98 cannot convert to 99. If Ru work as π acid and activate the triple bond, the next step should be nucleophilic attack to the triple bond. If the next step is oxidative addition to the amide bond, then Ru should coordinate with N or O first.
Answer: Thanks for your indication and we have made a correction in Scheme 29. Your inference is very reasonable and logical. However, the original reaction mechanism proposed by the authors began with the coordination of the ruthenium catalyst to the alkyne moiety and then the oxidative addition of ruthenium catalyst across the amide bond took place. As a review, we can’t change the reaction mechanism which was proposed by the authors. For your information, more details about the reaction mechanism could be found in the original literature (ref.42: Wu, C.-Y.; Hu, M.; Liu, Y.; Song, R.-J; Lei, Y.; Tang, B.-X.; Li, R.-J.; Li, J.-H. Ruthenium-catalyzed annulation of alkynes with amides via formyl translocation. Chem. Commun. 2012, 48, 3197-3199).
Comment 25: 21) Line 352 “, however” should be “. However”.
Answer: Thanks so much for your indication, and we have made the corresponding changes according to your suggestion.
Comment 26: 22) Line 368 “carbine” should be “carbene”.
Answer: Thanks so much for your indication, and we have made the corresponding changes according to your suggestion.
Comment 27: 23) Line 380 and 381 “depuration” should be “deauration”.
Answer: Thanks so much for your indication, and we have made the corresponding changes according to your suggestion.